# Field Use of Protective Bacteriophages against Pectinolytic Bacteria of Potato

**DOI:** 10.3390/microorganisms11030620

**Published:** 2023-02-28

**Authors:** Karel Petrzik, Josef Vacek, Martin Kmoch, Denisa Binderová, Sára Brázdová, Ondřej Lenz, Rudolf Ševčík

**Affiliations:** 1Institute of Molecular Biology, Biology Centre of the Czech Academy of Sciences, Branišovská 1160/31, 370 05 České Budějovice, Czech Republic; 2Department of Growing Technologies, Potato Research Institute Havlíčkův Brod, Dobrovského 2366, 580 01 Havlíčkův Brod, Czech Republic; 3Laboratory of Virology, Department of Genetic Resources, Potato Research Institute Havlíčkův Brod, Dobrovského 2366, 580 01 Havlíčkův Brod, Czech Republic; 4Institute of Food Preservation, Faculty of Food and Biochemical Technology, University of Chemistry and Technology (VŠCHT), Technická 3, 166 28 Prague, Czech Republic

**Keywords:** *Enterobacteriaceae*, *Solanum tuberosum*, *Limestonevirus*, biological control

## Abstract

The pectinolytic *Dickeya solani* bacterium is an important pathogen found in potatoes. We conducted laboratory and field experiments mimicking severe and mild *Dickeya* spp. infection and investigated the application of a mixture of two lytic bacteriophages before and after bacterial infection to protect the plants. Application of the phage solution to tuber disks and wounded tubers did not completely eliminate the infection but reduced the development of soft rot symptoms by 59.5–91.4%, depending on the phage concentration. In the field trial, plants treated with bacteriophages after severe *Dickeya* infection had 5–33% greater leaf cover and 4–16% greater tuber yield compared to untreated plants. When simulating a mild infection, leaf cover was 11–42% greater, and tuber yield was 25–31% greater compared to untreated plants. We conclude that the phage mixture has the potential to protect potatoes ecologically from *D. solani*.

## 1. Introduction

*Dickeya solani* is a relatively newly observed potato-infecting pectinolytic bacterium that has been seen in Europe since the beginning of the century and was established as a new species in 2014 [1]. Together with *Pectobacterium* spp., it is among the most devastating bacterial pathogens in agriculture, as they are the causal agents of bacterial blackleg and soft rot disease of potato tubers [2]. During infection, the bacteria produce characteristic extracellular pectinolytic enzymes that cause degradation of the plant cell wall, which manifests as typical tissue maceration. *D. solani* causes rapid top wilt of the growing potato plant, but the visible blackleg develops only in some varieties. Most losses in potato production caused by this bacteria are due to quality losses or seed rejection at certification [3]. Direct yield losses are estimated to be as much as 30–50% [4,5]. Further losses occur during storage and distribution to consumers, when the tissue of infected potato tubers softens quickly [6].

*D. solani* is a regulated non-quarantine pathogen under European Community law. It likely originates from a warmer environment and probably is unable to survive in the soil for more than 3 weeks in the absence of its host [3,7]. Therefore, *D. solani* is most commonly spread via latently infected seed tubers between fields and between growing seasons [3,8]. Subsequent foliar infection may be due to transmission by insects, splashing rainwater [9], or irrigation water [10]. Via infected plants, the bacteria can spread over long distances and live as epiphytes or facultative saprophytes in the soil and groundwater. Today, the genus *Dickeya* comprises twelve species: *Dickeya aquatica*, *D. chrysanthemi*, *D. dadantii*, *D. dianthicola*, *D. fangzhondgdai*, *D. lacustris*, *D. oryzae*, *D. parazeae*, *D. poaceiphila*, *D. solani*, *D. undicola*, and *D. zeae.* The symptoms caused by *D. solani* on potatoes are indistinguishable from those caused by other *Dickeya* spp. that also infect potatoes, i.e., *D. dadanthii* and *D. dianthicola* [11]. No effective chemical substances are currently available to control *Dickeya* spp. infection [12,13]; therefore, agricultural practices that can reduce bacterial contamination and accidental spread are applied. Cleaning and disinfection of machinery and equipment, monitoring of irrigation water, field inspection for blackleg symptoms, controlled multiplication of microtubers, post-harvest monitoring of seed stocks, and detection of the bacteria are used.

Most likely, no commercial potato cultivar is resistant to *D. solani* [2], but several lines of wild potato species, such as *Solanum microdontum*, have shown high levels of resistance to *Dickeya dianthicola* [14], as has *Solanum chacoense* M6 to *Pectobacterium* spp. [15]. A somatic hybrid of *Solanum tuberosum* and *Solanum brevidens* is also known for its high resistance to soft rot [16]. Nevertheless, the identification and introduction of the resistance genes [17] into commercial potato varieties will be laborious and complicated. Biological pest control based on antagonistic bacteria is a promising alternative or a complement to the integrative management strategy in potato cultivation. Antagonistic bacteria occupying the same niche can prevent the colonization of tubers by harmful bacteria and, thus, the development of disease symptoms. A mixture of bacterial antagonists containing *Enterobacter amnigenus*, *Rahnella aquatilis*, and three *Serratia* species was developed to protect potatoes from soft rot caused by *Pectobacterium* and *Dickeya* spp. and reduce the incidence of the disease by 46% [18].

On the other hand, the use of lytic bacteriophages (viruses in recent taxonomy) is an attractive and more targeted option for biological crop protection [12,19,20]. In general, phages show high specificity for bacterial hosts and are safe for other bacterial species. The stable coexistence of phage-resistant and phage-susceptible bacterial populations is thought to be established in nature [21]. Recently, more than 20 viruses infecting *Dickeya* spp. have been reported (17 from the *Ackermannviridae* family, 8 from the *Autographiviridae* family, and 5 unclassified tail viruses), but only the viruses of the *Ackermannviridae*, genus *Limestonevirus*, are widely distributed in Europe (GenBank accessed January 2023; [20]). Their use in biological crop protection has been investigated in several laboratory studies [22,23,24,25], and all showed promising initial results. Only one field trial was conducted [12], however, and it documented a yield increase of about 13% in cured plants compared to untreated tubers.

In this paper, we present the results of laboratory and field experiments simulating a very severe infection of plants with *Dickeya* and a nearly realistic mild infection depending on the treatment before and after infection with a mixture of φDs3CZ and φDs20CZ bacteriophages.

## 2. Materials and Methods

### 2.1. Bacteriophages and Bacterial Strains

Phages φDs3CZ and φDs20CZ were isolated from soils of potato-growing areas in the Czech Republic, propagated in *Dickeya* sp. CPPB-200 host (obtained from Crop Research Institute, Prague-Ruzyně), and purified as described by Petrzik et al. [26]. The stock suspensions of phages were maintained in TE buffer (10 mM Tris-HCl; pH 7.4; 10 mM MgSO_4_, 150 mM NaCl) at 10 °C. Phage concentration (PFU/mL) was determined by applying serially diluted phage to soft agar containing the *Dickeya* CPPB-200 host and observing the lytic zones. For bacterial concentration, the optical density OD_600_ = 1 was equivalent to 10^9^ CFU/mL [17]. 

### 2.2. Laboratory Tests

Potato cultivars from VESA a. s., a potato breeding company in Velhartice, Czech Republic, were used for sensitivity tests. In total, 17 potato varieties were tested, 3 varieties for industrial starch production and 14 all-purpose varieties. A total of 13 varieties were very early to early maturing varieties, and 4 varieties were semi-late or late maturing varieties (see Table 1). Disinfected potato tubers (n = 45) were cut into slices about 10 mm thick. Placed upon those disks were sterile 6 mm filter paper disks soaked with a bacterial suspension of 5 × 10^5^ bacteria/mL of *Dickeya* sp. CPPB-50 or *Pectobacterium carotovorum* CPPB-201. The slices were kept in a humid chamber at 21 °C in darkness and documented and evaluated 3 days post-inoculation (dpi). The percentage of macerated area was determined. 

The efficacy of the phage mixture at different concentrations against bacterial infection was simulated on potato slices of the Red Anna variety inoculated with the strain *Dickeya* sp. CPPB-50. Healthy tubers of the variety (tuber size 35–50 mm) were rinsed with tap water to remove adhering soil, disinfected with 1% sodium hypochlorite for 15 minutes, rinsed in sterile water, and then air-dried overnight at room temperature. Then, using a pipette, 200 µL of a phage solution consisting of 50% phage φDs3CZ and 50% phage φDs20CZ was dropped onto the potato slices at concentrations of 1 × 10^6^, 1 × 10^7^, and 1 × 10^8^ PFU/mL. The slices were left for 10 minutes to allow good diffusion of the solution into the tissue. Sterile 6-mm filter paper discs soaked with a bacterial suspension of 5 × 10^5^ bacteria/mL were then placed on the slices with tweezers. Treatment without the application of phages to the tuber slices (inoculation with bacterial suspension only) served as a positive control. Treatments with sterile water were used as a negative control. The slices were stored in a humid chamber and evaluated after 3 dpi. 

The order of application (bacteria first, phage second, then vice versa) was examined on whole tubers. The stem ends and bud areas of the tubers (n = 45) were punctured with a sterile steel stylus (diameter: 2 mm; length: 10 mm). Using a pipette, 10 µL of the phage solution at a concentration of 1 × 10^7^ PFUs/mL was dropped onto the wounds thus prepared. The tubers were left for 10 minutes to allow good diffusion of the solution into the tissue. The wounds were then infected with 10 µL of a bacterial suspension of 5 × 10^5^ CFU/mL using a pipette and sealed with parafilm. Conversely, the bacterial solution was first pipetted into the prepared wounds, and then the phage solution was applied. The tubers were placed on a filter paper soaked with distilled water (200 mL) in plastic boxes. The closed boxes were kept at 21 °C in the dark. Three days post-inoculation, the tubers were cut across the inoculation sites to measure the macerated area on the cross-section of the tubers (%) and to determine the extent of infection within each treatment. The macerated area on the cross-section of the tubers was evaluated using ImageJ 1.53e software (available at https://imagej.nih.gov/ij).

### 2.3. Field Trial

The field trial was established at the experimental site in Valečov in the Czech Moravian Highlands at an altitude of 460 m (49.639121N, 15.494967E). The soil type was cambisol, soil texture class was sandy loam with a proportion of 59.9% sand, 27.9 % silt, and 12.2% clay. Soil preparation in spring consisted of smoothing, application of mineral fertilizer by harrowing, preparation of the seedbed with the rotavator, and hand planting. The mineral fertilizer was applied according to the agrochemical soil analysis in a dosage of 115 kg N/ha (urea 46% N), potassium and magnesium in a dosage of 120 kg/ha as K_2_O and 40 kg/ha as MgO (Patenkali 30% K_2_O, 10% MgO). The total experimental area, including the paths for moving the tractor with the mounted sprayer, which separated the individual replications, was 1000 m^2^. The distance between the rows was 0.75 m, and the row length was 9.3 m. In one row, 32 tubers were planted manually at a distance of 0.3 m; the first and the last tubers were the protective ones. The cultivation plots for yield determination were three rows separated by protective rows, so that the yield of 90 hills was determined.

In the field trial, severe bacterial inoculation was carried out by cutting the tubers transversely and soaking them for 10 minutes in *Dickeya* spp. CPPB-050 bacterial suspension with 5 × 10^5^ CFU/mL. Mild bacterial inoculation was performed by merely puncturing the tubers (i.e., not cutting them transversely) and then soaking the punctured tubers in the same manner as described above. Thus, the potential for bacteria to enter the tubers through the much smaller damaged area was much lower (approximately 1000×). The damage to the tubers was, in every case, prepared on the day of planting. Phage suspension was administered either before or after bacterial inoculation by immersion in a phage solution consisting of 50% phage φDs3CZ and 50% phage φDs20CZ at a concentration of 10^7^ PFU/mL for 5 s. Bacterial inoculation of the intact potato without any phage treatment served as a positive control. All experiments were repeated three times with 90 potato plants in each combination. Sixty days after sowing, the experimental field was documented by EVO Lite+ drone’s (Autel Robotics, Shenzen, China) 20 Mpx camera, and the images were analyzed using ImageJ 1.53e software.

Statistical assessment of the experiments was carried out using variance analysis (one-factor ANOVA) and Tukey’s HSD test (*p* ˂ 0.01; STATISTICA 7 software, StatSoft, Tulsa, OK, USA).

### 2.4. Sequence Analyses

Nucleotide sequences of Dickeya limestoneviruses were obtained from GenBank and (Appendix A) were aligned using muscle [27] and mafft software by the FFT-NS-2 method [28], and analyzed by RDP5 with default setting [29]. Recombination events detected by at least 5 methods were considered.

## 3. Results

### 3.1. Potato Cultivar Sensitivity Test

Seventeen potato varieties produced by VESA and available on the Czech market, representing all-purpose potato varieties and starchy varieties for industrial use, as well as early and late maturing varieties, were tested for their susceptibility to *Dickeya* and *Pectobacterium* infections. All varieties showed some degree of susceptibility to the tested bacteria, and no variety showed minimal susceptibility (Table 1). The very popular variety Red Anna, a semi-late maturing all-purpose variety with medium-sized tubers, red skin, and yellow flesh, resistant to *Phytophthora infestans* and potato cyst nematodes, and showing high susceptibility to both bacteria (about 50% of area macerated), was selected for the field trials based on this result.

### 3.2. Virus Sequence Analysis

Dickeya viruses φDs3CZ and φDs20CZ had been isolated, sequenced, and characterized previously [26]. They had genomes of 155 285 bp and 154 720 bp, respectively, and were 99% identical. The φDs3CZ lysed the *Dickeya* CPPB-050 host strain faster than did φDs20CZ, however, and the latter had a 15% higher phage yield than φDs3CZ (Appendix A). Furthermore, both the viruses had extremely high content of mobile elements, as did the other Czech isolates. By closer sequence analysis using the RDP5 software, we found that all the Czech isolates of Dickeya limestoneviruses are recombinants with high probability (results supported by five RDP methods). The best example was φDs25CZ, which was identified as a recombinant of phage Coodle from Denmark as a major parent and phage PP35 from Russia as a minor parent (see Appendix A).

### 3.3. Laboratory Trial

The efficacy of the phages φDs3CZ and φDs20CZ was investigated on tuber slices inoculated with highly infective *Dickeya* CPPB-050 strain and evaluated as the decrease of macerated area in relation to the concentration of phages used in the treatment. As expected, the higher the concentration of phages used, the greater the observed decrease of macerated area (Figure 1, Appendix A). While the positive control had 80% macerated area, the concentration 10^8^ PFU/mL decreased that area to about 7%, the concentration 10^7^ PFU/mL to about 16%, and 10^6^ PFU/mL decreased the macerated area to about 33%. For practical reasons (especially limited capacity to produce highly concentrated phages for field trial), we performed our field tests using a phage concentration of 10^7^ PFU/mL. In the standard phage purification procedure with PEG/NaCl precipitation and subsequent centrifugation, we were able to prepare phages at concentrations as strong as 10^10^ PFU/mL.

In the next trial, the order of applying bacteria and phage was investigated. While the positive control had 88% macerated area, the arrangement wherein phage solution was applied first had a macerated area of just about 14%. In the arrangement which the bacterial infection was applied first, the macerated area was about 53% (Figure 2).

### 3.4. Field Trial

The field trial was prepared on an experimental field known as Valečov near Havlíčkův Brod in the Czech Republic’s Czech–Moravian Highlands. Planting was carried out on 28 April 2022, and harvest was done with a one-row potato bagging harvester on 30 September 2022. The sum of active temperatures above 10 °C (May–September) was 2390, and rainfall (April–September) was 461 mm, without irrigation. Three replicates simulating mild and severe infection of *Dickeya* CPPB-050 and pre- and post-inoculation phage treatment were planted with 90 tubers in each variant (Figure 3). Plant condition was evaluated as leaf coverage 6 weeks after planting, and tuber yield was compared.

Plants from tubers first inoculated with bacteriophages and followed by bacterial infection (mild or severe) were in all replicates in the best condition. Mildly infected plants showed as much as 42% greater leaf coverage compared with the (bacteria-only) positive control. In the case of severe bacterial infection, inoculated plants had leaf coverage as much as 33% greater compared to the untreated plants of the positive control (Figure 4). Plants from tubers that were first infected with bacteria and followed by bacteriophage treatment showed weaker positive effect from the treatment. They had 11% and 5% greater leaf coverage than the positive control in the case of mild and severe bacterial infection, respectively. 

Additionally, the tuber yield of mildly infected tubers was 25% to 31% higher than that of untreated plants and substantially to much higher than that of severely infected tubers. Phage treatment had a positive effect in both combinations (Figure 5).

## 4. Discussion

The efficient protection of commercial food production has become a major issue as agriculture is newly more oriented to sustainability and ecological concerns, and this demands new, modern, and environmentally friendly approaches. Even as *Dickeya* spp. can reduce potato tuber yields by as much as 25% [30], phages have the potential for specifically targeted and safe treatment against plant pathogenic bacteria [31]. Phages have a massive reproductive advantage over their bacterial hosts. After each infection, a phage produces more than 100 new particles, whereas a bacterial cell divides into two daughter cells. Consequently, phages quickly outcompete the bacterial host population but never completely destroy it [21]. Instead, a balance is established between their coevolving populations. In the case of *Dickeya* spp./Dickeya-specific bacteriophages, the limestoneviruses seem to offer the most promise in fighting these bacteria inasmuch as the pathogen occurs in many European countries and so do the corresponding bacteriophages. By contrast, *Aarhusviruses* and *Salmondviruses* have been localized only in limited areas. Moreover, *D. solani* has been isolated from geographically distant sites while indicating very limited genetic diversity within the species [25]. Dickeya limestoneviruses, on the other hand, have shown high sequence variability, which is due to the presence of numerous mobile elements that promote the occurrence of recombinant genomes [26]. Czajkowski et al. who collected nine Dickeya phages from Poland, also point out the high variability of their host range and protective effect, despite the strong similarity of genome fragment patterns [22]. We must bear in mind that the bacteriophages are constantly evolving, and so a protective bacteriophage mixture should be modified according to local conditions. To increase the maximum efficiency of the curative bacteriophage mixture, *Dickeya* strains of local origin were used for phage selection, and bacteriophages were selected from potato-growing areas in the Czech Republic.

The selected isolated lytic phages φDs3CZ and φDs20CZ are phages with the fastest lytic response to *Dickeya* infection and those with the most productive viral burst among the seven viruses found in the Czech Republic. The efficacy of the described phages on *D. solani* was studied in laboratory experiments using tuber slices and artificially injured whole tubers. Their efficiency in reducing potato tuber maceration (by at least 90%) is comparable to the effect of limestoneviruses ΦPD23.1 and ΦPD10.3, which reduce maceration by at least 80% [23]. In biological experiments with the phages φD1, φD2, φD3, φD4, φD5, φD7, φD9, φD10, and φD11, Czajkowski et al. [22] observed a 30–70% reduction in soft rot caused by *D. solani* on tuber slices compared to tuber slices inoculated with the pathogen alone. Phage treatment also has a positive effect on potato tubers even in the absence of target bacteria, possibly by eliminating indigenous soft rot bacteria [25]. It should be noted that in laboratory trials, a high concentration of *D. solani* and also high concentrations of bacteriophages are used to achieve a measurable infection and also visible protective effect within a reasonable time and at a suitable trial area. We assume that the natural infection is much weaker, and of course, only economically justifiable amounts of bacteriophages must be used in commercial production fields. The application of the effective bacteriophage concentration of 10^6^–10^7^ PFU/mL requires making several liters of highly concentrated bacteriophage stock solution per hectare. We increased phage yield from the bacterial lysate to 10^11^–10^12^ PFU/mL by filtration through the monolithic CIMmultus^TM^ QA column (Sartorius, Germany). That was two orders of magnitude higher phage yield per mL of culture compared to that achieved by the PEG/NaCl precipitation/centrifugation method. A second improvement was to modify the field applicator for bacteriophage application so that the bacteriophage suspension was sprayed directly onto the individual tubers from two lateral nozzles during planting, while a third nozzle applied the suspension directly to the soil surface (Figure 6). This arrangement saves a significant amount of bacterial suspension and increases the chance that phages will survive longer in the humid environment and interact with the bacterial host. In another study, when Czajkowski et al. inoculated φD5 phage directly onto the surface of potato tubers or into potting soil, no significant reduction in the number of phage particles was observed at 21 dpi [24].

Adriaenssens et al. who first isolated LIMEstone bacteriophage and then conducted a phage therapy trial with these phages, observed a strong efficacy of phage concentration and multiplicity of infection. The efficacy of the phages was dependent upon the concentration of the solution and the order of application. The application of phages to wounded tubers prior to bacterial inoculation resulted in a statistically significantly greater reduction in the development of disease symptoms compared to the application of phages after bacterial inoculation. In their field experiment with a concentration of 10^8^ CFU/mL of *D. solani* inoculum and subsequent treatment with a 10^10^ PFU/mL concentration of the phage, the treatment resulted in a 13% increase in tuber yield [12]. This is in agreement with our results where the yield increase was 4.0% with heavy bacterial infection and 31.9% with mild *D. solani* infection, followed by phage treatment. In a reciprocal experiment, where bacterial infection preceded phage inoculation, the yield increase was 16.2% and 25.9%, respectively, compared to untreated tubers in cases of severe and mild infection.

## 5. Conclusions

We can conclude that phages of the genus *Limestonevirus* isolated from local fields are effective against *Dickeya* spp., both in the laboratory and in field conditions. When the seed tubers are lightly infected, which is close to the agricultural practice of sowing certified tuber, it shows a significant protective effect on the health status of the plants and also on the tuber yield. Furthermore, the effectiveness of phages depends on the concentration of the treatment solution and the timing of its application. Application of phages prior to bacterial infection resulted in a significantly greater reduction in disease symptom development and overall plant health—measured as total leaf cover in the field—than the application of phages after bacterial infection. This result supports the statement that phage therapy is a good tool for biological pest control in crops.

## Figures and Tables

**Figure 1 microorganisms-11-00620-f001:**
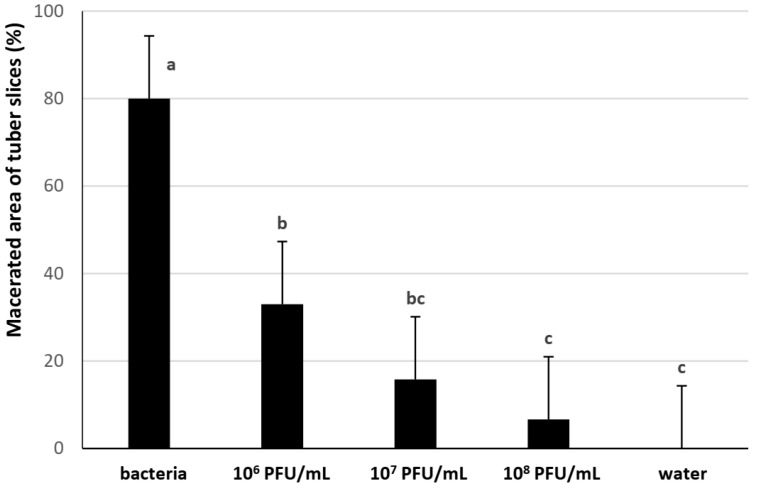
Effect of φDs3CZ and φDs20CZ mixture treatment on *D. solani* infection of tuber slices (% macerated area). Data are means of three independent replications, 45 tuber slices each. Letters over columns indicate statistically significant differences (*p* < 0.01) based on Tukey’s HSD test. The whiskers on columns indicate 0.99 confidence intervals.

**Figure 2 microorganisms-11-00620-f002:**
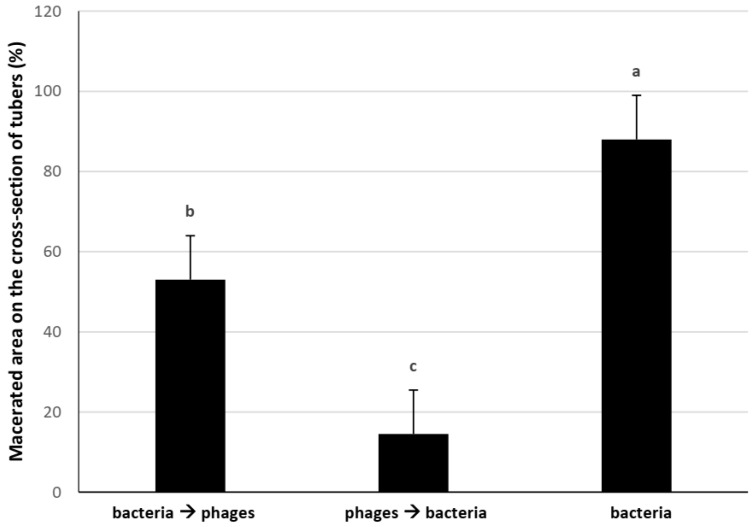
Effect of application order. Bacteria and phages were applied to whole potato tubers, and macerated area on the cross-section of tubers was measured at 3 dpi. Data are means from three replications, each consisting of 45 tubers. Letters over columns indicate statistically significant differences (*p* < 0.01) based on Tukey’s HSD test. The whiskers on columns indicate 0.99 confidence intervals.

**Figure 3 microorganisms-11-00620-f003:**
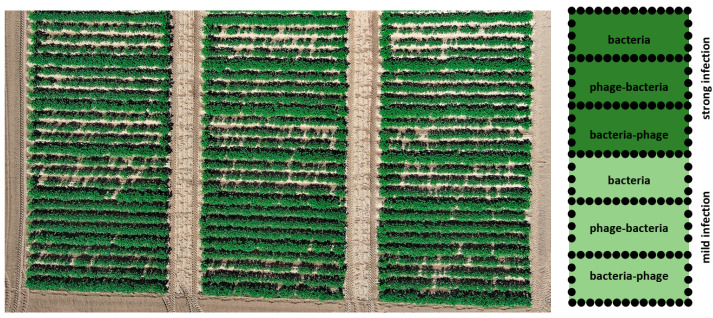
Field experiment. The 3 replications of mild and severe infection were separated by bare soil. Trials (3 rows each) were separated from one another by 1 row of the untreated cultivar Monika, and there was 1 plant of untreated Monika at the end of each row.

**Figure 4 microorganisms-11-00620-f004:**
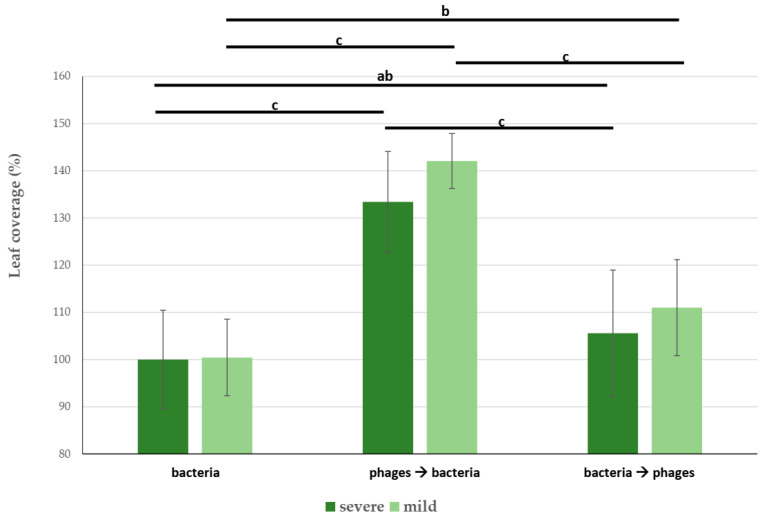
Leaf coverage of field trial measured from the drone. Bacterial infection alone, bacterial infection followed by phage application, and phage application followed by bacterial infection. Leaf coverage in the case of positive control (i.e., bacteria infection, no phage inoculation) was rebased to 100% for easier comparison. Field experiments were performed in 3 replications, 3 rows each. Leaf coverage was measured 6 weeks after sowing. Letters over columns indicate statistically significant differences (*p* < 0.01) based on Tukey’s HSD test. Whiskers indicate standard deviation.

**Figure 5 microorganisms-11-00620-f005:**
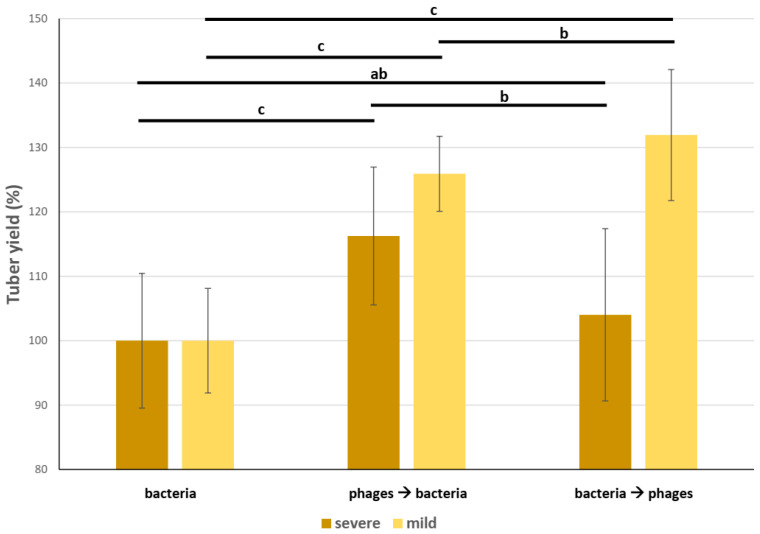
Tuber yield of field trial. Weight of tubers from 3 individual rows in 3 replications was measured. Bacterial infection alone, bacterial infection followed by phage application, and phage application followed by bacterial infection field experiments were performed in 3 replications, 3 rows each. Tuber yield in the case of positive control (i.e., bacteria infection, no phage inoculation) was rebased to 100% for easier comparison. Letters over columns indicate statistically significant differences (*p* < 0.01) based on Tukey’s HSD test. Whiskers represent standard deviation.

**Figure 6 microorganisms-11-00620-f006:**
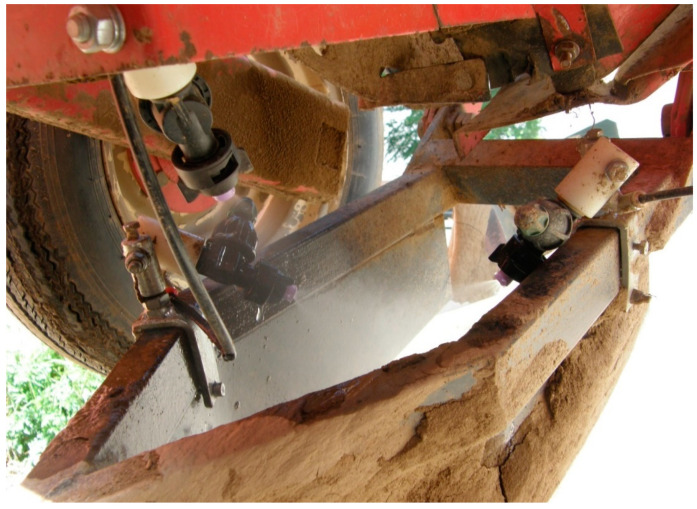
Detail of phage suspension spray application nozzles in the planting machine.

**Table 1 microorganisms-11-00620-t001:** Potato cultivar sensitivity to bacterial infection evaluated by potato slice inoculation.

Cultivar Sensitivity *	Bacteria Species	
	*Pectobacterium carotovorum*	*Dickeya solani*
9.00–7.5	-	-
7.49–6.00	Bella, Vysočina	Bella, Katy, Vysočina
5.99–4.50	●David^S^, Katy, Magda, Mariannka, Monika, Primarosa, Suzan	●David^S^, Dominika, Magda, Mariannka, Monika, Primarosa, ●Verne^S^, Suzan
4.49–3.00	Alice, Bohemia, Dominika, Jasmína, ●Jindra, Red Anna, ●Verne^S^, ●Westamyl^S^	Alice, Bohemia, Jasmína, ●Jindra, Red Anna, ●Westamyl^S^
2.99–1.00	-	-

* Sensitivity measured from 1 (highly sensitive, ≥75% macerated) to 9 (minimally sensitive, unmacerated). Sensitivity was evaluated at 3 dpi. ^S^ - in index-marked starchy cultivars. ● - marks semi-late and late maturing cultivars.

## Data Availability

Not applicable.

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
