# Peer review of "Field Use of Protective Bacteriophages against Pectinolytic Bacteria of Potato"

_microorganisms, 2023, doi:10.3390/microorganisms11030620_

Round 1
Reviewer 1 Report
In the manuscript "Field Use of Protective Bacteriophages against Pectinolytic Bacteria of Potato" K. Petrzik et. al. reported the field use of bacteriophages to control blackleg in potatoes.
The biological control of crop pathogens is the subject of intensive ongoing research as part of the premise of achieving sustainable and environmentally sound agricultural practices. In this regard, bacteriophages are natural enemies of bacterial pathogens exhibiting strong specificity and, for this reason, are promising tools for the biological control of phytopathogenic bacteria. The main question behind this conceptual approach is whether the technology can be adapted to large areas of cultivation, such as potato planting and pre-storage processing.
This paper describes an attempt to move from in vitro studies to a field experiment. Of course, this work is primary and further research is needed. Information about the novelty of the study compared to other works is not entirely clear and requires clarification in the text. Unfortunately, the work is not very relevant, although interesting data on the protective effect have been obtained, but this effect has been described for a long time. It is depressing that the authors did not resort to the use of formulations for the phage preparation, which might have helped to increase the effectiveness against the disease.
However, experiments conducted on potato slices, tubers and in the field show a satisfactory effect of phages against potato soft rot pathogens.
There are significant errors and issues in the work that require serious correction before publication in the journal.
• Entire text: carefully check italics, eg lines 158, 164, 141 and give all taxa names in the same form
• Line 14: Remove extra space
• Line 30: Throughout: sometimes spelled spp., sometimes sp. Please correct and make it the same for all references in the text.
• Line 36: Given that the previous sentence refers to D. solani, it makes more sense to write "this bacteria"
• Line 56: better to write "phytosanitary analysis"
• Lines 66-69: the proposal should be moved to the discussion section
• Line 78: Theirs?
• Line 83: Dickeya have to on italic
• Line 86: For ease of reading and familiarization with the article, this section should be divided into subsections (eg, bacterial strains, lab experiments, field experiments and other.
• Line 92: Give a brief description of the varieties (maturing group, morphological features that may have contributed to differences in susceptibility to the bacterium, and are these varieties used in your country, what share of the tuber market for seed purposes do they occupy)?
• Line 94: How did you check the concentration of bacteria and phage before the experiment, please describe in more detail.
• Line 95: In what containers and how were the slices incubated? In a room or maybe in a thermostat with active ventilation? Please describe in more detail
• Line 96-97: What method was used for analysis? What is the number of repetitions and what were the options?
• Line 99-100: Why was this bacterium-variety pair chosen?
• Line 103-105: Justify WHY these proportions were taken and what does "30-50%" mean? Does it mean 30, 50 or something in between? Give exact values or ratio
• Line 104-105: Why were these particular concentrations taken? Justify
• Line 114: PFUs/mL : delete "s"
• Line 120: After 3 dpi
• Line 124: write manufacturer and country
• Line 125: Since this section is devoted to field trials, please add information about the experimental plot: total area, area of plots, presence of protective rows, soil type and characteristics (including granulometric composition), climatic features (sum of active temperatures. rainfall, availability of nutrients in the soil, and presence/absence of irrigation. It is also necessary to add to this section information on the method of planting potatoes, the width and length of row spacing, and soil preparation measures for the experimental plot.
• Line 127: "bacteria/mL" change on "CFU/mL"
• Line 136: What variety was used and why?
• Line 137: On what equipment and how was it documented? write more
• Line 140: write country and manufacturer
• Line 153 and table 1: provide statistical processing of the received Initial data
• Table 1: In the discussion section, write the reason for choosing these particular varieties (country distribution/preliminary studies or random selection?
• Drawing 3: Improve drawing quality:
column "bacteria" - move the letter a to the front or below the standard deviation indicator.
10^7 PFU/ml - set the standard deviation indicator exactly. It is not clear why we need gray vertical 3-5 lines in the columns.
• 10^6 PFU/ml - change to 106 PFU/ml
• Line 189: replace where with "where in"
• Figure 4: improve the quality of the drawing, remarks are the same as
• Line 201: Rewrite the title of the drawing. For example: Drone Image of Field Tool Options....
• Line 208-209: "Plant condition was evaluated" replaced with "Plants conditions were evaluated"
• Figure 6: Plot the results of the statistical processing in a graph.
• Figure 7: Where are the results of the statistical analysis of this experiment?
• Line 251: Aarhusviruses and Salmondviruses should be in italics.
• Line 311: Authors should add a paragraph on proposed ways of addressing the problem of low efficiency of the phage preparation and proposed research on this issue. Unfortunately, in the discussion there are no references to works that describe possible preparative forms of phage preparations for use in crop production.
• Line 320: change the laboratory and in agriculture to in the laboratory and field conditions
• Line 320: change seed to tubers
• Supplementary Figure 2: Convert values in vertical axis to 10X PFU/mL or Log PFU/mL
• Supplementary Figure 2: In the Methods and Results section, provide information about the programs and algorithms used in the recombinant assay, and in the results, briefly describe the results obtained from this assay.
• Line 460: In the notes to the table, describe the abbreviations "nt" and "AC number

Reviewer 2 Report
In the manuscript “Field Use of Protective Bacteriophages against Pectinolytic Bacteria of Potato”, the authors proved that the phage mixture has potential to protect potato ecologically from D. solani through potato sensitivity test and field experiment. They further demonstrated that the efficacy of the phages was dependent upon the concentration of the solution and the order of application. To put the results to practical use, they improved the phage yield and modified the field applicator for bacteriophage application. This manuscript experiment basically repeated the previous experiment, supported the previous research results, and made a new discovery on this basis. Whereas, new findings in this study lack novelty, and the research on phage therapy in agricultural practice is not comprehensive and indepth. I afraid the novelty and significance of this study still have the distance to the demand of Microorganisms.
The author's experimental results can support his conclusion. But I think there could be more richness in the experimental design. For example, they obtained the higher the concentration of phage used, the smaller the area of softening observed by using three different concentrations of phage mixtures. But the concentration gradient is limited. We know that application of the phage solution to tuber disks and wounded tubers did not completely eliminate the infection. Is it possible to find an optimal value of concentration by setting more concentration gradients and comparing the quality and yield of potato under different concentration of phage mixtures? This study is only limited to this region, and the results have little significance for other regions, so the research can be carried out in different dimensions (region, potato varieties, bacterial varieties, etc.). In addition, the conclusion is too superficial, as if retelling previous research results (Adriaenssens et al. 2012) without summarizing new research results, which can be supplemented.
Other comments:
1, Line 270) Revise “et al”.
2, Move Figure 5 after this paragraph “The field trial was prepared … 6 weeks after planting and tuber yield was compared.”
3, Figure 7 should be moved to line 292.
4, Line 270) Here should not be converted to the next line.
5, Line 408) Change “32(5)” to italic.
6, Line 424) Place “2019,” after “Plant Dis.” and change “103” to italic.
Round 2
Reviewer 2 Report
The new version is fine for me.